



# Central Himalayan tree-ring isotopes reveal increasing regional heterogeneity and enhancement in ice-mass loss since the 1960s

Nilendu Singh[1], Mayank Shekhar[2], Jayendra Singh[3], Anil K Gupta[4], Achim Bräuning[5*], Christoph Mayr[5], and Mohit Singhal[1]

[1]Centre for Glaciology, Wadia Institute of Himalayan Geology, Dehradun 248001, India
[2]Birbal Sahni Institute of Palaeosciences, Lucknow 226007, India
[3]Wadia Institute of Himalayan Geology, Dehradun 248001, India
[4]Center for Oceans, Rivers, Atmosphere and Land Sciences, IIT Kharagpur 721302, India
[5]Institute of Geography, University of Erlangen-Nuremberg, Erlangen 91058, Germany

*Correspondence to*: Achim Bräuning (achim.braeuning@fau.de)

**Abstract.** Tree-ring $\delta^{18}O$ values are a sensitive proxy of regional physical climate, while their $\delta^{13}C$ values are a strong predictor of local ecohydrology. Utilizing available ice-core and tree-ring $\delta^{18}O$ records from central Himalaya (CH), we show an increase in east-west heterogeneity since the 1960s. Further, $\delta^{13}C$ records from transitional western glacier valleys provide a robust basis of reconstruction of about three centuries of glacier mass balance (GMB) dynamics. Annually resolved GMB is based on regionally-dominant and diverse plant-functional species since the 1743 CE. Results indicate three major phases: positive GMB up to the mid-nineteenth century, the middle phase of slightly negative but stable GMB, and an exponential ice-mass loss since the 1960s. Reasons of accelerated mass loss are largely attributed to anthropogenic climate change, including concurrent alterations in atmospheric circulations (weakening of the westerlies and Arabian Branch of the Indian summer monsoon). CH-scale, multi-decadal isotopic and climate coherency analyses specify an eastward declining influence of westerlies in this monsoon-dominated region. Besides, our study provides a long-term context for recent GMB variability, which is essential for reliable projection and attribution.

## 1   Introduction

Glaciers in the Himalaya-Tibet orogen are an important component of the regional hydrological cycle, and a major fraction of regional potable water is stored and provided by them. However, recent shifts in climate dynamics have imposed a serious alteration in the equilibrium of these glaciers (Bandyopadhyay et al., 2019; Bolch et al., 2012; Maurer et al., 2019; Mölg et al., 2014; Yao et al., 2012; Zemp et al., 2019). High uncertainty prevails in future projections, as a sound understanding of glacier fluctuations and its response to climate change on a longer timescale is completely lacking (e.g., erroneous statement in the Fourth Assessment Report of the IPCC, Kargel et al., 2011). Reliable projections of future Himalayan ice mass loss require robust observations of glacier response to past and ongoing climate change. Long-term estimation of glacier mass balance (GMB) is also imperative for regional water security. Currently, coupled glacier-climate models even do not agree on the sign of change and hence projections on GMB is ambiguous (Watanabe et al., 2019, Jury et al., 2019; DCCC, 2018).



Nevertheless, a consistent picture emerges of net ice mass loss in recent decades, which is highest in the Western and Central Himalaya (except Karakoram/Pamir) (Bolch et al., 2012; Brun et al., 2017; Dehecq et al., 2019; Maurer et al., 2019; Mölg et al., 2014; Shekhar et al., 2017; Yao et al., 2012).

The central Himalayan glaciers show a rather homogeneous behaviour (Azam et al., 2018; Bandyopadhyay et al., 2019; Brun et al., 2017; Dehecq et al., 2019; Kääb et al., 2012; Sakai and Fujita, 2017; Yao et al., 2012). In this study, we focus on the transitional climate zone between the Western and Central Himalaya, where knowledge about long-term glacier dynamics in relation to climate change is unknown. Evidences from tree-ring isotopes and hydroclimatic studies suggest that the Himalayan-scale heterogeneity in glacier mass balance behaviour is primarily determined by the conjoint effect of the winter

westerlies (WD) and Indian summer monsoon (ISM) (Fig. 1 and references therein). The influence of the ISM declines towards the northwest Himalaya, and the westerlies progressively becomes dominant. Towards the eastern Himalaya, the influence of the Arabian Sea (AS) branch of the ISM declines, and the Bay of Bengal branch (BoB) dominantly regulates the climate, besides having influence of the East Asian monsoon (Benn and Owen, 1998; Bookhagen and Burbank, 2010; Hochreuther et al., 2016; Liu et al., 2014; Lyu et al., 2019; Mölg et al., 2014; Sano et al., 2017; Yang et al., 2008; Yao et al.,

2012). Monsoon-influenced glaciers, particularly those in the transitional climatic zone (such as western region of the central Himalaya) are particularly sensitive to climate warming than winter-accumulation type glaciers (Kargel et al., 2011; Sakai and Fujita, 2017). Moreover, a weakening in the moisture delivery systems (i.e., WD and ISM) since the mid-twentieth century had a direct impact on the summer-accumulation type glaciers found in the region (Dahe et al., 2000; Hunt et al., 2019; Joswiak et al., 2013; Khan et al., 2019; Roxy et al., 2015, 2017; Sano et al., 2012, 2013, 2017; Singh et al., 2019; Xu

et al., 2018; Yadav, 2011).

In the present study, we attempt to reconstruct glacier mass balance history since the end of the Little Ice Age (since 1743 AD) in the transition zone between the ISM and the westerly-dominated climate in the central-western Himalaya (Fig. 1). We use tree-ring stable isotope ($\delta^{13}$C and $\delta^{18}$O) chronologies of three dominant tree species of two different plant functional types and synthesize available regional $\delta^{18}$O chronologies from different archives, including tree-rings and ice-cores across

the central Himalaya (Fig. 1; Table S1).

Field-based mass balance measurements are logistically challenging in the Himalaya. Nevertheless, since the 1980s, workers have endured monitoring of four benchmark glaciers in the Indian central Himalaya (Fig. 1). Based on a detailed analysis of hypsometric curves and several morphological and glaciological factors of these four valley glaciers (see materials and methods; Bandyopadhyay et al., 2019; Pratap et al., 2015; Garg et al., 2017, 2019), and building upon previous studies

(Shekhar et al., 2017; Bandyopadhyay et al., 2019; Azam et al., 2018), we compiled a mean observed glacier mass balance time-series since the 1980s (1982 - 2013) (Fig. 1, Fig. S1, Table S2).

The length of observed mass balance records principally limits our understanding of the response of glaciers in the Himalaya to climate change. Remote sensing techniques provide the best alternative in this regard (Garg et al., 2017, 2019; Maurer et al., 2019; Kääb et al., 2012; Dehecq et al., 2019; Brun et al., 2017; Bandyopadhyay et al., 2019; DCCC, 2018). Nevertheless,

observation-based mass balance record cannot be extended beyond a few decades. To understand long-term changes, we





must rely on proxy data. One of the valuable proxies is ice-core isotopes, which is yet to be obtained from the Indian Himalaya. Tree-rings are another reliable and sensitive proxy for the reconstruction of glacier history and have been widely used in mountainous valley regions around the world (Bräuning, 2006; Duan et al., 2013; Gou et al., 2006; Hochreuther et al., 2015; Larocque and Smith, 2005; Linderholm et al., 2007; Nicolussi and Patzelt, 1996; Solomina et al., 2016; Tomkins et
al., 2008; Watson and Luckman, 2004; Xu et al., 2012; Zhang et al., 2019). Tree-ring width (TRW) is known to bear the influence of non-climatic factors like biological tree age that overlaps climatic signals (and sensitivity issues such as temperature, precipitation and topography; Bunn et al., 2019; Zhang et al., 2019). TRW records of Himalayan evergreen conifer trees (such as *Abies* spp., *Cedrus* spp., *Picea* spp.) have been widely used to reconstruct hydroclimate and natural hazards (Borgaonkar et al., 2009, 2018; Cook et al., 2003; Singh et al., 2006; Singh and Yadav, 2013; Yadav and Bhutiyani,
2013; Yadav et al., 2011; Yadav, 2011), but there is only one TRW-derived glacier mass balance reconstruction (Shekhar et al., 2017). However, tree-ring stable isotopes are a reliable source of past hydroclimate variability because of their sensitivity to local/regional climate, the coherence in the climatic response, and the well understood control by the environmental conditions regulating tree physiology (Levesque et al., 2019, Sano et al., 2012, 2013, 2017; Singh et al., 2019, Zeng et al., 2017). Although many studies reconstructed hydroclimate in the Himalaya using tree-ring isotopes (e.g., Sano et al., 2012,
2013, 2017; Singh et al., 2019; Xu et al., 2018), glacier mass balance reconstructions from precise tree-ring isotope series are still not available. In this study, we have utilized tree-ring carbon isotope ($\delta^{13}$C) chronologies of three different species belonging to two plant functional types (PFT) that differ in their annual phenological cycle (Table S1). One PFT includes evergreen conifers (*Abies pindrow* and *Picea smithiana*) that have been widely used to reconstruct hydroclimatic regimes across the Himalaya and are abundantly found in the moist valley regions of the high central-western Himalaya. Moreover,
we have investigated the dendroclimatological potential of a major dominant species (*Aesculus indica*) that belongs to the PFT of broadleaf deciduous trees, as its growth period (April to September) coincides with the warm-wet phase during the summer monsoon season (ISM) in the central Himalaya (and lacks westerlies signal during complete winter dormancy). Additionally, we have utilized available Dasuopu ice-core and tree-ring isotope chronologies from different locations of the central Himalaya to substantiate our results (Fig. 1). This study aims (i) to identify the tree-ring parameter (TRW and/or
$\delta^{13}$C, $\delta^{18}$O) of deciduous and evergreen tree species containing the strongest mass balance signature, (ii) to reconstruct annual mass balance using the identified best proxy chronology, and (iii) to analyse the reconstructed mass balance in relation with regional climate, local forcing factors, and large-scale atmospheric circulation.

## 2   Materials and Methods

### 2.1 Study region and climate

The present study focuses on the glaciers of the Ganga basin (Uttarakhand, India) in the Indian central Himalaya. The study region extends from latitude 30.15° to 31.03° N and longitude 78.78° to 80.73° E. The highly glaciated upper reaches of the Ganga basin encompass about 1000 glaciers of varying size, which cover an area about 2850 km$^2$ (Raina and Srivastava,



2008; Bandyopadhyay et al., 2019). Previous studies found that glaciers in the region are mainly fed by the ISM. Therefore, glaciers in the region are usually classified as summer-accumulation type glaciers (Azam et al., 2018; Sakai and Fujita, 2017). Snow accumulation during winter (December–March) is influenced by a precipitation regime driven by mid-latitude westerlies (WD). Decadal-scale meteorological records from the study region are available only for lower elevations (< 2000 m asl) (Fig. 1). Therefore, long-term gridded temperature and precipitation datasets were obtained through the Climatic Research Unit (CRU TS.3.22, 0.5 latitude x 0.5 longitudes, 1901-2015) (Harris et al., 2014). Analyses of available meteorological records indicate that mean annual precipitation is ~800 mm, of which the warm-wet summer months (April-September) receive about 80%. Mean annual temperature varies around 4°C, with a minimum (-7°C) in January and maximum (13.5°C) in June (Fig. S2). As previously mentioned, studies indicate an increasing trend in temperature, while precipitation during both ISM and WD seasons show a recent declining trend. As local climate is the common factor influencing both glacier mass balance and physiology or growth of the trees growing in the valleys, we first analyzed tree growth-climate relations (Fig. S3). Averaged monthly temperature and total monthly precipitation over a period spanning from October of the preceding year to September of the current growth year were correlated with $\delta^{13}C$ chronologies of the species (as $\delta^{13}C$ is strong predictor of local climatology) (Fig. S3). In addition, various climatological indices (SST, ENSO, ENSO-Modoki, PDO, and IOD) were obtained from the archive (http://climexp.knmi.nl) to analyse large-scale climatic relations.

## 2.2 Glacier mass balance data

Four benchmark valley glaciers that are distributed across the Ganga basin namely Dokriani (DOK), Chorabari (CHO), Tipra Bamak (TIP) and Dunagiri (DUN) have been individually monitored for their mass balance over the time-period from 1982 to 2013 (with a few gap years, Fig. 1; Table S2). Two glaciers (DOK and CHO) lie in the Garhwal Himalaya, while two (DUN and TIP) are located in the Kumaun Himalaya (India). We used the available glaciological mass balance records to produce best possible time-series (Table S2). We built a mean mass balance time-series based upon previous studies (Shekhar et al., 2017; Bandyopadhyay et al., 2019; Borgoankar et al., 2009) and the premises that the glaciers show a homogeneous behaviour to changes in the regional climate and that their mass balances show a high inter-correlation (Azam et al., 2018). Given the same climatic forcing owing to the location in the humid central Himalaya and the similar geomorphological characteristics (Garg et al., 2017, 2019) (Table S2), the four valley glaciers should generally correlate well (Solomina et al., 2016). Ideally, land-terminated valley type glaciers of a simple configuration and hypsometry are most suitable to infer paleoclimatic information (Solomina et al., 2016). The four studied glaciers of the Ganga basin are of this kind and are therefore designated by workers as 'benchmark glaciers'. The glaciers are of similar small size (except DUN), have simple shapes and moderate elevation ranges, and thus their recent deglaciation responded to changes in the climate equivalently (Table S2). A surface ice velocity study (Garg et al., 2019) (Table S2) confirmed a similar response-time of these glaciers. Geomorphological features determining glacier hypsometry largely regulate the response of an individual glacier to changes in the climate. Thus, we prepared hypsometric curves of these four benchmark glaciers (Fig. S1); which

were convex for the DOK and CHO glaciers, while they were concave for the TIP and DUN glaciers. These curves and other similar glacio-morphological indices (Garg et al., 2017, 2019) (Table S2) provide a reasonable basis to compile these individual mass balance time-series (Shekhar et al., 2017; Bandyopadhyay et al., 2019).

**Figure 1.** Study region in the western-central Himalaya (WCH) at transitional climate zone encompassing four benchmark glaciers of the Ganga basin (lower panel; A: Dokriani Glacier, DOK; B: Chorabari Glacier, CHO; C: Tipra Bamak; D: Dunagiri Glacier) and indicating tree-ring sampling sites, meteorological and aethalometer stations. Published tree-ring isotope sites in WCH include: (1) Manali (Sano et al., 2017), (2) Uttarakashi (Singh et al., 2019), (3) Jageswar (Xu et al., 2018). Published tree-ring isotope sites from monsoon-influenced central-eastern Himalaya (ECH) and southeast Tibetan Plateau include: (4) Humla (Sano et al., 2011), (5) Ganesh (Xu et al., 2018), (6) Bhutan (Sano et al., 2013) and Reting (Grießinger et al. 2011), Xinpu (Wernicke et al., 2017), Ranwu (Liu et al., 2014), respectively. Central Himalaya 'Dasuopu Ice-core' has been indicated as blue dotted circle (Thompson et al., 2000).



### 2.3 Tree-ring data

The three tree species utilized in this study are ubiquitously distributed species throughout the central-western Himalaya have been used earlier to reconstruct past climatic variations. The collected species occupy both aspects of a glacier valley, stretching from treeline to down-slope towards the valley. About 20 to 30 increment core samples of healthy tree individuals (to minimize the influence of non-climatic factor on growth) were collected from higher elevation sites (2500 - 3800 m asl) from two representative glacial valleys (Dokriani: DOK and Pindar: PIN) encompassing the basin in the Indian central Himalaya (Fig. 1). Tree-ring isotopic signals (particularly $\delta^{18}O$) are coherent over a very large area, as shown by high inter-species and inter-site correlations in the Himalaya (Sano et al., 2012, 2013, 2017; Xu et al., 2018; Grießinger et al., 2019; Table S1). Recently, Singh et al. (2019) reconstructed regional ISM precipitation derived from $\delta^{18}O$ chronologies of the same three tree species (two PFTs) from the DOK valley and similarly found high inter-species and inter-site correlations. Due to unavailability of long-term tree-ring stable isotope chronology from PIN valley, we have utilized $\delta^{13}C$ chronologies of three tree species from the DOK valley.

In summary, we adopted a methodology to reconstruct glacier mass balance history from tree-rings successfully tested before (Nicolussi and Patzelt, 1996; Duan et al., 2013; Zhang et al., 2019). We used standard dendrochronological methods and techniques to develop tree-ring width (TRW) and stable isotope chronologies. Then, we calibrated the climate response using linear regression models, and verified the reliability of our reconstruction. The leave-one-out cross-validation method (LOOCV; Michaelsen, 1987; Yadav and Bhutiyani, 2013; Duan et al., 2013; Zhang et al., 2019) was used to verify our reconstruction, given the relative shortness of glacier mass balance data (23 years: 1982-2013 after omitting gap years of 1990-1992, 1996, 1997, 2001-2003 and 2011) (Table S3).

### 2.3.1 Tree-ring width chronology development

Two core samples per tree were collected at breast height using 5.15 mm diameter increment corers. Standard dendrochronological procedures (Fritts, 1976; Holmes, 1983) such as mounting and surface smoothing were applied to render the ring boundaries clearly visible (this process was labour-intensive for deciduous tree species (*Aesculus indica*) relative to the conifers). The ring widths of the samples were measured at a resolution of 0.001 mm using a LINTAB™ system interfaced with a computer. Cross-dating was performed by matching variations in ring-widths in all cores to determine the absolute age of each ring. Dating and ring-width measurement quality control was conducted using the COFECHA computer program. Coherence in growth pattern of trees of the species and ring-width plots revealed a common regional climate signal affecting growth of the trees. However, TRW chronologies of individual species could not be calibrated with compiled glacier mass balance data due to the existence of a weak correlation.



### 2.3.2 Stable isotope chronology

A weak correlation between TRW and glacier mass balance could arise due to sensitivity issues (Bunn et al., 2019). At high-elevations in a valley environment, temperature and precipitation signals mix and make TRW records difficult to interpret (Bunn et al., 2019). In view of this, we resorted to tree-ring isotopes ($\delta^{13}C$) of the species that are known to be climate-sensitive. Interestingly, we found a strong correlation only between $\delta^{13}C$ of our studied conifer species and compiled glacier mass balance (*Abies pindrow*: $r = 0.596$, $P < 0.001$, n = 23 and *Picea smithiana*: $r = 0.631$, $P < 0.001$, n = 23). The

correlations are strong enough to establish a significant calibration model (Shah et al., 2013; Shekhar et al., 2017; Yadav and Bhutiyani, 2013; Vehtari et al., 2017; Zhang et al., 2019). However, a weak correlation was obtained with deciduous species (*Aesculus indica*: $r = 0.343$, $P = 0.1101$, n = 23). This could be due to lack of storage of winter-time climate record, as the species remains physiologically active only during wet-warm period from April to September.

Five trees per species were selected for stable isotope analyses based on best TRW inter-series matches. Each year's growth-

rings were dissected with precaution with a sharp scalpel under the microscope. To remove possible juvenile effect, the innermost approximately 30-40 rings of each tree core were omitted from the analyses. Wood samples were grounded using an ultracentrifuge mill (Retsch ZM1). Isolation of cellulose from whole wood and carbon isotope analysis were carried out at the Institute of Geography, Erlangen, Germany. Cellulose was extracted from the wood samples using the method of Wieloch et al. (2011). Isolated cellulose was homogenized using an ultrasonic method as described in Laumer et al. (2009).

Thereafter, the homogenized samples were freeze-dried. Before the pooling procedure, we checked co-relatedness in all five individual time series at 20-year intervals in the entire chronology. About 270 μg of cellulose was weighed into tin capsules with a micro balance (ME36S, Sartorius, Germany). The carbon isotope analyses were performed with an elemental analyzer (NC 2500, Carlo Erba, Italy) linked to an isotope-ratio-mass spectrometer (IRMS; DeltaPlus, Thermo-Finnigan, Germany). Prior to isotope analyses, samples were thoroughly dried in a vacuum drying cabinet at 60°C. Isotope values were calibrated

with international (IAEA-CH7, USGS-41) and laboratory standards (peptone). The analytical precision was equal or better than 0.2 ‰. Carbon ($\delta^{13}C$) values of samples were calculated by comparison with isotope ratio-predetermined peptone and cellulose lab standards and certified international isotope standards (IAEA-601, IAEA-602, IAEA-CH7, USGS-41), which were inserted frequently in the course of sample measurements. Isotope ratios are presented in the common $\delta$ notation against PDB as:

$$\delta^{13}C = \left[ \frac{\left(\frac{^{13}C}{^{12}C}\right)_{sample}}{\left(\frac{^{13}C}{^{12}C}\right)_{PDB}} - 1 \right] \times 1000 \ (‰)$$

The final tree-ring carbon isotope chronology was corrected for the incorporation of isotopically light carbon released by the burning of fossil fuels and increasing $CO_2$ concentration as proposed by McCarroll and Loader (2004) and McCarroll et al. (2009). The correction procedure applied here has the advantage of being objective, as it effectively removes any declining trend in the $\delta^{13}C$ series post 1850 CE, which is attributed to physiological response to increased atmospheric $CO_2$

concentrations (McCarroll et al., 2009).



## 2. 4 Statistical analyses

Tree growth–climate relationship was analysed using simple Pearson correlation analysis. The relationship with glacier mass balance was analysed using the pair-plot correlation package 'Performance Analytics' (Carl et al., 2010) in *R* (Table S4, S5, Fig. S3). Correlations were computed between monthly temperature and rainfall data and the tree-ring $\delta^{13}C$ chronology for a window from the previous year's October to the current year's September.

To better visualize the comparison between our reconstructed glacier mass balance time-series and other hydroclimatic reconstruction series from the monsoon dominated Central Himalayan region (Fig. 1), data time-series were standardized using Z-scores and smoothed with 11-year fast Fourier transform to highlight common climate signals.

Based on correlation analysis, a linear regression model (Briffa and Jones, 1990, Cook et al., 1994) was used to perform the reconstruction using LM module in *R* (ggplot2 package, Wickham, 2016). The leave-one-out cross-validation method (LOOCV; Michaelsen, 1987) was used for the entire calibration period (1982–2013) and to verify the reconstruction (Table S3). This method is most suitable when the length of observed records is short (Shah et al., 2013; Shekhar et al., 2017; Yadav and Bhutiyani, 2013; Vehtari et al., 2017; Zhang et al., 2019). In this method, each observation is successively withdrawn; a model is estimated on the remaining observations, and a prediction is made for the omitted observation. The LOOCV analysis was performed using package 'caret' (Kuhn et al., 2015). Rigorous statistics including sign test, the reduction of error (RE), and the correlation coefficients were calculated to evaluate the similarity between observed and estimated values. Sign test measure the degree of association between two series by counting the number of agreements and disagreements. The series are highly correlated if the number of similarities is significantly larger than the number of dissimilarities. The RE statistic provides a rigorous test of the association between actual and estimated series. Positive value indicates the predictive capability of the model. A positive RE is an evidence of a valid regression model (Fritts, 1976). In addition, other rigorous statistics, viz., root mean square error, coefficient of efficiency (CE), and Durbin–Watson (DW) test were carried out to evaluate the linear regression model (Table S3).

Spatial and temporal correlations (moving correlations) were used to identify the coherence between reconstructed mass balance and gridded (0.5°×0.5°) temperature, precipitation, SST, ENSO, ENSO-Modoki, PDO, and IOD for the studied region in the central Himalaya. The variability of the climate and glacier mass balance reconstruction in the frequency domain was investigated using the multi-taper method (MTM) of spectral analysis (Mann and Lees, 1996) and wavelet transform (Grinsted et al., 2004) to identify periodicities and their temporal variability in the reconstructed data.

## 3  Results and discussion

### 3.1 Annual glacier mass balance reconstruction

The detailed results on analysed relationship between $\delta^{13}C$ and compiled glacier mass balance as well as with available climate datasets are presented in Table S4 and S5. Descriptive statistics of $\delta^{13}C$ chronologies and inter-species correlation





(for common period) are shown in Table S1. The mean difference (2.0 - 3.0 ‰) between $\delta^{13}$C time-series of two PFTs indicates a higher level of isotope discrimination in the broadleaf species relative to conifers. Nevertheless, a significant and positive inter-species correlation exists between them. This indicates an influence of common and coherent climatic factors

on the physiological processes. Climate-response functions indicate that for both PFTs, correlation strength between $\delta^{13}$C and mean annual temperature (MAT) or mean annual precipitation (MAP) remained similar (MAT; *Abies pindrow*: $r = -0.215$, $P < 0.05$, n = 66; *Aesculus indica*: $r = -0.281$, $P < 0.05$, n = 66; *Picea smithiana*: $r = -0.405$, $P < 0.001$, n = 66), (MAP; *Abies pindrow*: $r = 0.243$, $P < 0.05$, n = 66; *Aesculus indica*: $r = 0.115$, $P < 0.05$, n = 66; *Picea smithiana*: $r = 0.185$, $P < 0.05$, n = 66). Results on monthly and seasonal climate-response function analyses are provided in supplementary

material (Fig. S3, Table S5).

Correlation analyses between observed annual mass balance and $\delta^{13}$C of two PFTs indicated strongest correlation with the evergreen conifers (*Abies pindrow*: $r = 0.596$, $P < 0.001$, n = 23 and *Picea smithiana*: $r = 0.631$, $P < 0.001$, n = 23). In contrast, correlation with the deciduous species (*Aesculus indica*: $r = 0.343$, $P = 0.1101$, n = 23) is weak during summer (April-September) and non-significant during winter. Therefore, we utilized both evergreen conifer species for our

reconstruction (Table S4). A linear regression model was employed for the reconstruction of annual glacier mass balance (GMB) over the past 273 years and the corresponding equation is:

GMB $= 8.4526 + 0.3839 * \delta^{13}C_{conifers}$

Where, $\delta^{13}C_{conifers}$ is the mean chronology of *Abies pindrow* and *Picea smithiana* and GMB is annual glacier mass balance (meter water equivalent; m w.e.). The detailed model statistics are presented in Table S3. Model and calibration–verification

statistics indicate the reliability and strength of our reconstruction model (Table S3; Fig. S4; Fritts, 1976; Vehtari et al., 2017). Validation tests including the number of sign agreements between the reconstructed series and observed mass balance records, and cross-correlation between reconstruction and measurements are significant (Fig. S4). However, the error estimates are based on measured mass balance data of only 23 years (1982–2013, with gap of few years), so a possibility of uncertainty still exists in the reconstruction, particularly during the pre-observation period. Nevertheless, the use of sensitive

isotope chronology ($\delta^{13}$C) and the combination of two conifer species in this study may help to minimize several factors responsible for higher uncertainty, such as sensitivity issues, decreasing sample size and temporal variability, which is unavoidable with TRW.

### 3.2 Three major phases in the mass balance dynamics

Historical records of glacier change are rare from the Himalaya. However, some studies provide evidence that the monsoon-

influenced southeast Tibetan Plateau (TP) glaciers and those of the central Himalaya have responded synchronously to the change in climate. In contrast, glaciers in the western regions of the Himalaya-Tibet orogen behaved asynchronously (Solomina et al., 2016; Owen et al., 2008; Kaspari et al., 2008; Liu et al., 2013; Hochreuther et al., 2015; Xu et al., 2012; Bräuning et al., 2006). Studies highlight the relative importance of two moisture delivery systems (westerlies and Asian summer monsoon) in driving regional glacier fluctuations; how variations in moisture delivery systems have changed at



millennial to glacial-interglacial time-scale and their impact is even perceptible at interannual to decadal timescale (Hou et al., 2017, Mölg et al., 2014).

In a gross regional sense, the current state-of-the-knowledge suggests that since the last glacial advance (LIA) glaciers are in a general state of retreat (1850s), whereby the role of regional climate in regulating regional glacier fluctuations has increased gradually. Therefore, across the Himalaya this period (since the LIA) is characterized by dominancy of retreat,

advance, and/or standstill regimes (Mayewski and Jeschke, 1979; Mayewski et al., 1980; Bolch et al., 2012; Rowan, 2016; Solomina et al., 2016). Available records from the central Himalaya indicate a state of glacier retreat since the 1850s, regardless of the glacier type (Mayewski and Jeschke, 1979), with a recent acceleration in ice mass loss as indicated by remote sensing studies (Maurer et al., 2019; Bandyopadhyay et al., 2019).

Our 273 year long GMB reconstruction agrees with the general retreat of glaciers since the mid-19th century (Fig. 2).

Smoothing of the reconstruction with an 11-year moving average and break-point analyses revealed three distinguishable main phases: (1) a phase of positive mass balance up to ~1870 CE, coincident with LIA in the central Himalaya; (2) reorganisation of Northern Hemisphere atmospheric circulation following LIA and progressively increasing influence of regional climate could have resulted in a phase of near zero or stable mass balance that lasted up to 1960s and (3) a phase of accelerated ice mass loss since the 1960s. This phase corresponds to a global glacial retreat and can be attributed to

increasing temperatures, combined with a decline in ISM and westerly circulations, and anthropogenic climate change (e.g., Bollasina et al., 2011).

Hydroclimatic evidences such as regional and composite tree-ring $\delta^{18}O$ chronologies from the monsoon-influenced region (Fig. 1: from southeast TP to Central Himalaya) (An et al., 2014; Singh et al., 2019; Grießinger et al., 2011; Wernicke et al., 2017; Liu et al., 2014; Sano et al., 2012, 2013, 2017; Shi et al., 2012; Xu et al., 2018) and speleothems and Dasuopu ice-core

record from Central Himalaya (Thompson et al., 2000; Kaspari et al., 2008; Denniston et al., 2000; Kotlia et al., 2012; Kotlia et al., 2015; Liang et al., 2015) show that regional climate changed since the 1860s, with a reorganisation of hemispheric atmospheric circulation. Consequently, regional hydroclimate abruptly shifted towards a drier phase concurrent with the changes in atmospheric circulation. These records show a spatially coherent signal and serve as a validation test of the accuracy of our GMB reconstruction (Fig. 2). Particularly, tree-ring and speleothem $\delta^{18}O$ records (Singh et al., 2019, Sano et

al., 2012, 2013, 2017; Xu et al., 2018; Kotlia et al., 2015; Liang et al., 2015) from our studied region (Fig. 1) indicate increased westerly precipitation prior to 1860-70 CE (Murari et al., 2014; Yang et al., 2008) that resulted in a phase of positive mass balance. Increased winter snowfall could be anticipated from a southward shift of south-westerly winds due to reduced Northern Hemisphere air temperatures during the LIA (Rowen, 2016). The stronger influence of mid-latitude westerlies in driving glacier variability in monsoonal high Asia (Mölg et al., 2014) may have led to higher snowfall and

positive mass balance prior to 1870s as observed in our reconstruction (Fig. 2). Regional temperature reconstructions (Zhu et al., 2011, Borgaonkar et al., 2018, Yadav et al., 2011) also suggest a cold and cloudy climate prior to 1850, which was followed by warmer and sunnier climate thereafter (Liu et al., 2014; Xu et al., 2012). Indeed, a brief negative trend in mass balance during the 1770s to 1790s could be due to warm winter temperatures (Huang et al., 2018) and below-average snow





accumulation during mega-droughts as observed around this period (Cook et al., 2010; Thompson et al., 2000; Kaspari et al.,
2008) (Fig. 2). The close correspondence between reconstructed mass balance and regional hydroclimate reconstructions
supports the notion of hemispheric synchronicity to climate change prior to 1860-70 CE (Solomina et al., 2016). However,
gradual changes in regional oscillations in temperature and moisture delivery sources in the Himalaya-Tibet orogen,
particularly a change in the interplay of ISM and westerlies could have induced regionally distinguished climate zones with
specific behaviour of glacier dynamics (section 3.3).

Glacial history during the late-Holocene and LIA in our studied region (Murari et al., 2014), the above mentioned proxy
records and climate dynamics studies (Mölg et al., 2014; Khan et al., 2019) indicate an out-of-phase relationship between the
westerlies and ISM influence. Therefore, regardless of the strength of the westerlies (Joswiak et al., 2013), ISM weakening
since the 1860s tended to shift the regional hydroclimate towards a drier and a warmer climate. It could have favoured ice
mass loss that led to a slightly negative mean mass balance (-0.046 ± 0.134 m w.e., ±SD) during middle phase (1870-1959)
in our reconstruction. Timing of our reconstructed glacier advancement or recession is consistent with records of cold
periods during the early 19[th] (1815–1825) and 20[th] century (1900–1910) (Zech et al., 2003). A brief period of declining mass
balance from 1865 to 1885 could be ascribed to warm winter temperatures observed during 1848 to 1894 (Huang et al.,
2018), and the late Victorian Great Drought during 1875-1878 (Singh et al., 2018; Cook et al., 2010). Slightly mean negative
but stable mass balance observed from 1920 to 1960 is consistent with reported mass balance from the Himalaya (Bolch et
al., 2012) and pluvial condition from enhanced ISM activity during 1920 to 1960 (Sinha et al., 2015).

Severe glacial mass loss (-0.437 ± 0.191 m w.e.) since the 1960s corresponds to a further and prominent shift towards drier
conditions arising out of the weakening of both the ISM and westerlies (Dahe et al., 2000; Hunt et al., 2019; Basistha et al.,
2008, Singh et al., 2019, Sano et al., 2012, 2013, 2017). Increasing regional and global temperatures accompanied with
progressive regional industrialization may have exacerbated the retreat rates. We observed a doubling of average ice mass
loss rate (-0.577 ± 0.022 m w.e. year$^{-1}$) (±SE) in the last thirty years compared to 1960-1985 (-0.275 ± 0.022 m w.e. year$^{-1}$).
An extensive remote sensing based study encompassing the last forty years observed a consistent and similar trend of glacier
loss across the Himalayan transect (Maurer et al., 2019). For our studied region, Bandyopadhyay et al. (2019) computed
mean weighted mass balance as 0.61 ± 0.04 m w.e. year$^{-1}$ (2000-2014), while we observed 0.65 ± 0.02 m w.e. year$^{-1}$
(excluding very large-sized Gangotri Glacier). Similarly, several geodetic mass balance estimates from the western-central
Himalayan region are comparable to our result and are within the error limits of earlier studies (DCCC, 2018;
Bandyopadhyay et al., 2019).

In recent decades with decreasing intensity of moisture influx (ISM and WD), the importance of regional temperature in
regulating mass balance behaviour is gradually increasing (Fig. 3). Several studies from the eastern Himalaya-Tibet orogen
have emphasised on increased influence of regional temperature in determining mass balance behaviour (Bräuning, 2006;
Yang et al., 2008). Our results confirm historical observations made in the early 19[th] century (without availability of
geochronological data to constrain the timing) that the central Himalayan glaciers have been receding since the 1850 CE
(Mayewski and Jeschke, 1979; Rowen, 2016). The rate of ice mass loss accelerated since the 1960s and almost doubled in





the last thirty years with respect to pre-1985. Here, our result contradicts the notion of UNEP (2009) that 'despite the widespread shrinkage of the Himalayan glaciers in area and thickness, the nature of shrinkage has not changed significantly over the last 100 years' (UNEP, 2009).

**Figure 2**. Recontructed mass balance for the studied region (WCH) since 1743 AD (coloured inset indicates annual and pentadal (5 years) mass-change rate since the 1960s with std. errors). Lower panels show its comparison with annnual snow acculmulation derived from Dasuopu Ice Core from central eastern Himalaya (Thompson et al., 2000) and available long-term records from central Himalaya, India (Bolch et al., 2012) (lower panel). Dark lines indicate 11-year moving averages. Horizontal dashed lines are results of break-point analyses that reveal three major phases in the reconstruction.





### 3.3 Increasing regional heterogeneity

Substantial evidences exist from the monsoon-influenced Himalaya-Tibet orogen (Fig. 1), which suggest that glacial fluctuations were more or less synchronous during the LIA prior to 1850s (Hochreuther et al., 2015; Bräuning, 2006; Yang et al., 2008; Xu et al., 2012; Mayewski and Jeschke, 1979; Rowen, 2016, Owen et al., 2008). Following the LIA, a global

readjustment in atmospheric circulation resulted in a southward shift of Inter Tropical Convergence Zone (ITCZ). It broadly affected the Asian summer monsoon and the region progressively moved towards a drier phase, subsequent to a breakdown in ITCZ and Northern Hemisphere temperature relation since the mid-nineteenth century (Reidely et al., 2015; Xu et al., 2012). Nevertheless, the strength of westerlies in the region remained unaffected till the mid-twentieth century (Joswaik et al., 2013; Hunt et al., 2019; Khan et al., 2019). Thus, depending upon the geographical location and proximity to the oceans,

a distinct regionality was induced with decline in summer monsoon precipitation. The middle phase (mid-nineteenth to mid-twentieth century) in our reconstructed GMB and annual snow accumulation recorded in Dasuopu Ice Core (DIC) from the central Himalaya support this point of view (Fig. 2). During this period, cryospheric mass balance turned slightly negative, with a gradual decline in summer precipitation, which is more prominent towards the western part (Fig. 2, 3).

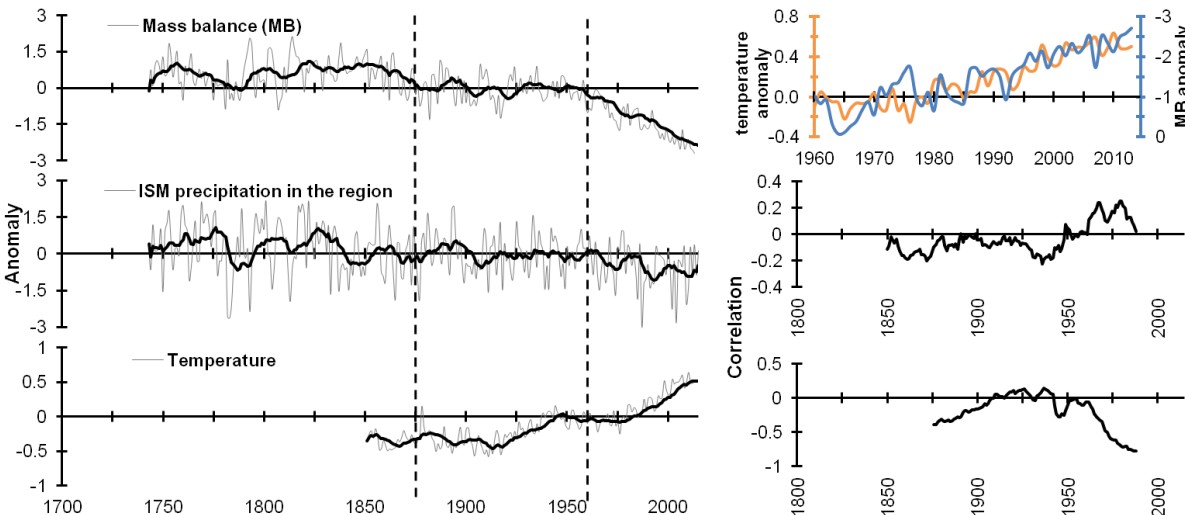

**Figure 3.** (a) Left panel indicates coherence in anomalies of mass balance with summer monsoon precipitation and temperature. Dark lines indicate 11-year moving averages. Right middle and lower panels indicate 51-year running correlations of mass balance with precipitation (Singh et al., 2019) and temperature (HadCRUT4), respectively. Upper right panel indicates enhanced correlation between mass balance and temperature after the 1960s and the effect of the global warming hiatus of the late 1990s on mass balance.





**Figure 3 (b).** Plot indicates anomalies in reconstructed annual mass balance and regional tree-ring δ¹⁸O series (Manali: Sano et al., 2017; Uttarakashi: Singh et al., 2019; Jageswar: Xu et al., 2018; Humla: Sano et al., 2012; Ganesh: Xu et al., 2018; Bhutan: Sano et al., 2013). Dark lines denote 21-year moving averages. Corresponding right panels indicate low frequency temporal correlations (51-year running correlations) with reconstructed mass balance. Different behaviour of regional chronologies, particularly a phase shift in WCH chronologies (Manali, Uttarakashi and Jageswar) is prominent after the mid-twentieth century.

Dendroglaciological and paleoclimate studies suggest that on a centennial timescale temperature changes remain the prime

factor for glacier fluctuations rather than precipitation changes (Yang et al., 2008; Wang et al., 2019). After the mid-twentieth century (1960s), the role of temperature increased in determining mass balance behaviour with further decline in moisture influx both from ISM and WD (Roxy et al., 2015, 2017; Dahe et al., 2000; Hunt et al., 2019; Basistha et al., 2008, Singh et al., 2019, Sano et al., 2012, 2013, 2017). Correlation analysis with CRU temperature data indicate a high coherence



with mass balance dynamics after the 1960s (r = -0.78, $P < 0.001$). Interestingly, reconstructed GMB even sensitively

corresponds to the effect of slowdown in global temperature increase since the late 1990s (warming hiatus) (Fig. 3a). In contrast, correlations with gridded precipitation including that of northern India and ISM rainfall indicate a low association. However, we observed a high coherence (at both low and high frequencies) with local $\delta^{18}O$ reconstructed summer monsoon precipitation derived from several regionally dominant tree species (Singh et al., 2019) (Fig. 3a). Moreover, we also found a tight correspondence with regional (Fig. 1) tree-ring $\delta^{18}O$ chronologies; the strength of which declined towards the eastern

central Himalaya and Bhutan (Fig. 3b, Table S6). Results indicate an abrupt change in hydroclimate after the mid-twentieth century, which is particularly prominent in western central Himalaya. Moving correlations (51-year) between mass balance and regional tree-ring $\delta^{18}O$ chronologies indicated a phase shift in the western part, which is more prominent relative to the eastern central Himalaya after the 1960s (Fig. 3b).

Sufficient reliable and sensitive records are available those exemplify regional hydroclimatic heterogeneity and show that the

drivers of glacier fluctuations in the western part of the central Himalaya (WCH) are somewhat different from the eastern central Himalaya (ECH) (Fig. 3b). Here, we contend that caution should be applied when referring the glaciers in WCH as summer-accumulation type glacier. We showed that over the WCH (compared to ECH), the westerlies still have a significant impact on annual mass balance behaviour. Earlier studies confirm that glaciers in the ECH and further east are mainly fed by summer monsoon precipitation and thus are undisputedly classified as summer-accumulation type (Sakai and Fujita, 2017).

However, our results reveal a contrasting hydroclimate relation between WCH and ECH, with an opposite behaviour of moving correlation patterns (51-year) between records of regional glacier/snow accumulation and tree-ring $\delta^{18}O$ series (Fig. 4). When comparing the $\delta^{18}O$ chronology of the deciduous species (*Aesculus indica*) growing in WCH, we find correlation pattern is strikingly similar to ECH, rather than to WCH. In contrast to conifer species that take up snowmelt water containing winter precipitation isotopic signal in the early growing season (Huang et al., 2019); while, the broadleaf

deciduous species takes up soil water mostly after leaf flush and full canopy development in May. At this time of the year, the snowmelt signal may already be gone, so the species does not reflect the winter precipitation signal. These results support our interpretation that the winter westerlies have a strong impact on annual mass balance of glaciers in WCH compared to ECH. The correlation patterns indicate an abrupt phase change in WCH since the 1960s, while correlation in ECH remains stable, suggesting a super-positioning role of temperature and differential precipitation seasonality derived

from the westerlies and the two branches of the ISM (AS and BoB) (Fig. 2, 4).

Analyses of tree-ring $\delta^{18}O$ records and water stable isotope ratios from the central Himalaya specify greater influence of the Arabian Sea branch of the ISM in WCH, the strength of which declines towards the eastern Himalaya. While, the climate in ECH is dominantly regulated by the Bay of Bengal branch, along with some influence of East Asian summer monsoon (Sano et al., 2013; 2017; Sinha et al., 2015; Liu et al., 2014). At the northwestern periphery of ISM incursions in WCH, moisture

flux from the Arabian Sea competes with the influence from Bay of Bengal (Sano et al., 2017). Thus, the role of the Arabian Sea branch increases in WCH, particularly when the Bay of Bengal branch is relatively weak (Sano et al., 2017).



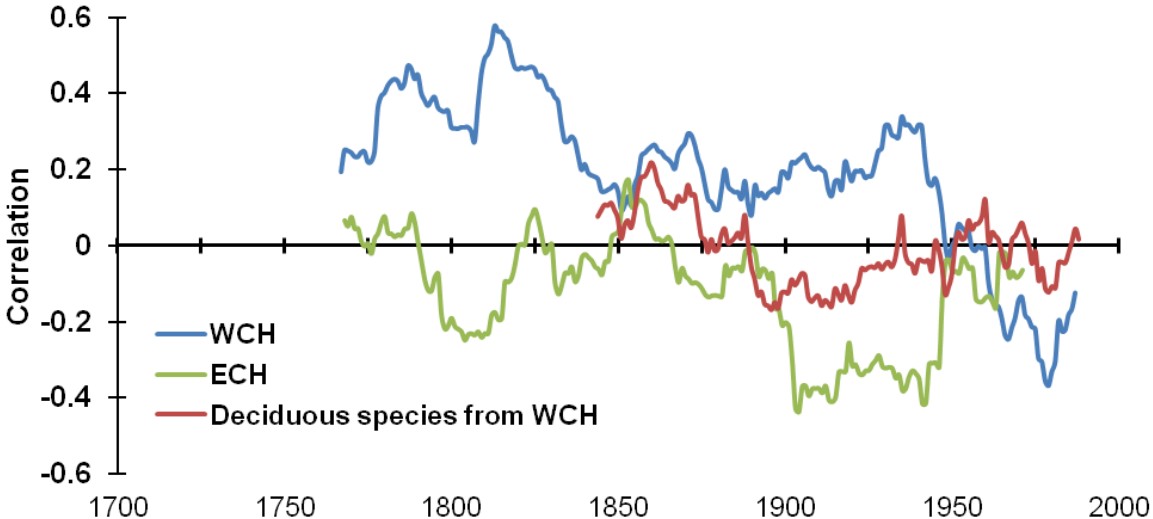

**Figure 4. Low frequency correlations between regional cryospheric dynamics and hydroclimate**. Blue line indicates 51-year moving correlations between reconstructed glacier mass balance (GMB) and regional tree-ring $\delta^{18}O$ (Manali, Uttarakashi and Jageswar) from the western central Himalaya (WCH). The abrupt phase shift since the 1960s is peculiar for WCH. Green line indicates 51-year moving correlations in the eastern central Himalaya (ECH) between annual snow accumulation from Dasuopu ice-core (Thompson et al., 2000) and regional $\delta^{18}O$ (Ganesh and Bhutan). While, the red line indicates 51-year moving correlations between GMB and tree-ring $\delta^{18}O$ of a dominant deciduous species (*Aesculus indica*) growing in WCH. Its similarity in the coherence with that of ECH is remarkable and illustrative of the summer monsoon influence during the annual growing period (April-September).

It is pertinent here to mention about differential influence and behaviour of the two branches of the ISM. Although the Arabian Sea (AS) and the Bay of Bengal (BoB) are part of Indian Ocean, but crucial ocean-atmosphere interaction

experiments have shown that they possess distinctly different features (Saikranthi et al., 2019). They are entirely different in terms of sea surface temperature (SST) and background atmosphere and related precipitating systems. The monsoonal winds and low-level Findlater jet are stronger over the AS than BoB. Tropospheric thermal inversions are more frequent and stronger over the AS than BoB. The variability in SST is larger over the AS than over the BoB. The SST in the AS cools between 10° and 20 ° N during the monsoon season, whereas warming is observed in all oceans between the same latitudes

(Misra et al., 2019; Saikranthi et al., 2019; Roxy et al., 2015). Thus, AS plays a predominant role in regulating the ISM rainfall variability, which appears to be particularly impactful for prevailing climate in the WCH.

Compelling evidences suggest an increased warming over the Indian Ocean since the mid-twentieth century. However, warming over the AS is monotonous for more than a century, at a rate faster than in any other region of the tropical oceans (Roxy et al., 2015, 2017; Misra et al., 2019). Currently, the AS is the largest contributor to the trend in global mean SST.

The abnormal warming over the AS tends to weaken the land-ocean thermal contrast and it has been implicated for a weakening of the ISM since the mid-twentieth century (Roxy et al., 2015, 2017). Besides, warming over the AS has also been shown to reduce the magnitude of the westerlies, as their interaction enhances the moisture convergence over the AS, leading to decreased precipitation from the westerlies (Misra et al., 2019). This mechanism may explain the decline in precipitation frequency, snowfall, and total precipitation amount from the westerlies since the mid-twentieth century (Hunt et





al., 2019; Das et al., 2002; Shekhar et al., 2010, Kumar et al., 2015, Khan et al., 2019). Moreover, a general decline in the

frequency of the westerlies has been attributed to the widening and weakening of the subtropical jet, while decreasing

meridional wind shear and mid-tropospheric baroclinic vorticity tendency has also been implicated (Hunt et al., 2019; Peings

et al., 2019). Besides, upper tropospheric tropical warming and Arctic amplification do not oppose, but rather concur to

weaken the subtropical jet (Hunt et al., 2019; Peings et al., 2019). Tree-ring $\delta^{18}$O records from Bhutan to the southeast TP

(Sano et al., 2013; Hochreuther et al. 2016; Lyu et al. 2019) show more or less unaltered conditions during the 20th century.

Thus, weakening of the BoB branch as indicated in some studies (Roxy et al., 2017) needs reappraisal. However, irrespective

of the strength or variability of the Bay of Bengal branch, accelerated ice mass loss since 1960s could be possibly ascribed to

the combined effect of a decline in moisture influx form the Arabian Sea and the westerlies.

Given the importance of the westerlies (Mölg et al., 2014) and its weakening, as reflected in the decline of the indices of the

Arctic Oscillation and North Atlantic Oscillation (Hunt et al., 2019; Peings et al., 2019), we focus on El Nino-Southern

Oscillation (ENSO), the Indian Ocean dipole (IOD), and the Pacific Decadal Oscillation (PDO), which are all known to

heavily modulate the summer monsoon precipitation. Hydroclimatic studies from the monsoon-dominated Himalaya-Tibet

orogen indicate a pervasive but temporally unstable coherence. Low frequency coherence exists between tree-ring $\delta^{18}$O

derived regional hydroclimatic reconstruction and the above indices of atmosphere-ocean interaction that appear to

strengthen in recent decades (Sano et al., 2012, 2013, 2017; Singh et al., 2019; Hochreuther et al. 2016; Lyu et al. 2019). In

accordance, our results too show a strong correlation at multi-decadal time-scale between tree-ring $\delta^{18}$O chronologies

available from WCH and the above indices (Fig. 5).

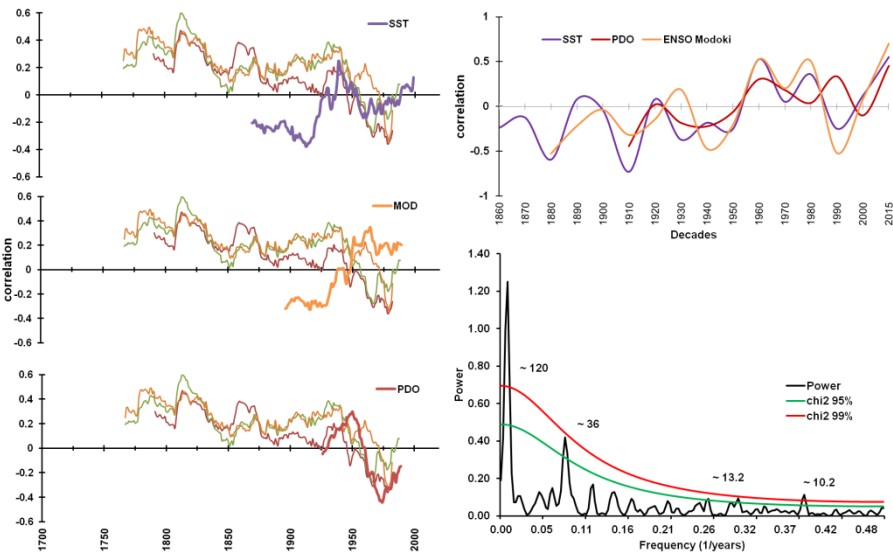

**Figure 5.** Left panel shows the temporal correlation behaviour at multi-decadal time-scale (51 years) between tree-ring $\delta^{18}$O chronologies available from the western central Himalaya (Manali: Sano et al., 2017; Uttarakashi: Singh et al., 2019; Jageswar: Xu et al., 2018) and Nino4 SST (violet line), ENSO Modoki (orange line), and Pacific Decadal Oscillation (PDO: red line). Upper right panel indicates strengthening decadal correlations between mass balance and these indices after the 1960s. Lower right panel is the plot of spectral analysis of mass balance time-series which reflects low frequency variations preserved in the reconstruction.



Tree-ring δ¹⁸O chronologies from the monsoon-dominated Himalaya generally contain a 2-5 years cyclic signal which is coherent with the ENSO cycle (Xu et al., 2018, Singh et al., 2019; Sano et al., 2012, 2013, 2017). However, spectral analysis

of our GMB time-series showed significant periodicities ($P < 0.05$) at decadal and multi-decadal scales, which reflect low frequency variations preserved in the reconstruction (Fig. 5). Lack of high frequency periodicities indicates a minor role of ENSO in modulating glacial hydroclimate on interannual or shorter timescale. Low periodicity was also found in a regional reconstruction of mass balance (Shekhar et al., 2017). Cross-correlations between reconstructed GMB and Nino4 SSTs revealed a weak negative correlation ($r = -0.25$, $P < 0.05$, 1901-2015). Interestingly, weak and non-significant negative

correlations between δ¹⁸O based summer monsoon rainfall reconstruction for our study region and SST (Singh et al., 2019) during 1909-1926 and 1957-2015 coincides with a negative mass balance trend (Fig. 2). While, significantly strong negative correlations observed during 1870-1908 and 1927-1956 coincide with positive mass balance trends (Fig. 2). This indicates a clear influence of SSTs on mass balance behaviour particularly for the WCH at low frequency timescale. Wavelet coherence analyses of reconstructed GMB with annual Nino4 SST revealed unstable common frequency composition in the different

series over time, indicating a wobbly linkage with the change in SSTs (Fig. S5), as observed in previous studies of regional hydroclimate (Sano et al., 2012, 2013, 2017; Singh et al., 2019; Hochreuther et al. 2016; Lyu et al. 2019). Similarly, we noted a weak correlation with other indices (Fig. S5). However, decadal correlations between mass balance and these indices strengthen after the 1960s (Fig. 5), indicating a negative influence of monotonous ocean warming on mass balance dynamics through precipitation feedback.

A lack of strong coherence in the high frequency bands between GMB dynamics and SST indices is possible because once the monsoon circulation sets in the region, the local weather begins to decouple from the circulation. There is complete absence of monsoon footprint on regional glaciers both from atmospheric drivers and from mass-balance perspectives (Mölg et al., 2014). Point-scale micrometeorological experimentation and surface energy balance analyses of a typical glacier from WCH (Pindari Glacier, Fig 1) also indicate a tight land-atmosphere coupling during the summer- as well as in the winter-

accumulation seasons. The coupling turned negative and remained weak only during the transition phases (pre- and post-monsoon), making it amenable to atmospheric perturbations (Singh et al., 2019). Thus, any repeated atmospheric alterations during the summer monsoon season such as active or break periods may not be able to strongly influence the mass balance dynamics. However, critical transition phases, particularly the pre-monsoon melt season has the potential to shape mass balance on decadal or even longer timescale (Mölg et al., 2014; Singh et al., 2019).

The pre-monsoon season is also important for the onset of summer accumulation season (during ISM). Numerical experiments and simulations in Himalaya-Tibet orogen have identified seasonal changes during the pre-monsoon that lead to the onset of ISM. The plateau maintains a large-scale thermally driven vertical circulation, which initially remains unconnected from the monsoon circulation. Rising motion exists only on the far western side of the Plateau during winter. Gradually, it extends towards the eastern side as the pre-monsoon season progresses. The ISM onset over the region is thus

an interaction process between the Plateau-induced circulation and northward migrating rain-bearing depression from AS and BoB. It is thus highly possible that warming-induced decline in winter-spring snow cover particularly over the western



TP could alter this interaction process and monsoon onset. The weakening of the relationship between snow cover over the western TP and ISM rainfall could be a possible manifestation (Zhang et al., 2019). Studies have affirmed that atmospheric conditions during pre-monsoon are pivotal in determining mass balance even at the decadal scale or longer timescale (Mölg et al., 2014). These studies also suggest anti-correlated precipitation variability between ISM and mid-latitude circulation during the pre-monsoon season. The interaction between these large-scale circulations in this region is largely determined by the position of thermally driven subtropical jet streams (Peings et al., 2019; Hunt et al., 2019). Thus, a possibility emerges that anthropogenic factors influencing tropospheric warming could alter the position of jet streams and annual mass balance behaviour, particularly in the transitional climate zone in the WCH.

## 3.4 Local forcing factors

Our region has experienced an enhanced warming trend during pre-monsoon season since the 1960s that has amplified in the last 2-3 decades (Lau et al., 2010; Gautam et al., 2010; Xu et al., 2009; Gautam et al., 2009, Prasad et al., 2009). This is most likely due to increased aerosol loading, enhanced atmospheric water vapour content (Mortin et al., 2016), and anthropogenic greenhouse ($CO_2$) emissions. These factors increase radiative heating rates, aid to net atmospheric warming and create a positive feedback loop to tropospheric warming. Enhanced tropospheric warming has the potential to alter the position of subtropical westerly jet and as described earlier, determines ISM strength by influencing the AS branch and ultimately the mass balance.

Of the local factors influencing tropospheric heating, greenhouse forcing of $CO_2$ emissions is pervasive. Studies suggest an inverse geological relation between glaciation events and atmospheric $CO_2$ concentration (Willeit et al., 2019; Macdonald et al., 2019). Causes of enhanced $CO_2$ concentration over the Himalaya include deforestation, increased frequency of forest fires, regional industrialization, and emission and transport of greenhouse gases. Beside major contribution from anthropogenic activities, sub-glacial carbonate weathering, especially sulphide oxidation process coupled to carbonate carbonation is a potential source of $CO_2$ to the atmosphere. Fast kinetics associated with these processes in glacial environment is also responsible for present day atmospheric $CO_2$ concentration (Shukla et al., 2019). The influence of rising atmospheric $CO_2$ concentrations on $\delta^{13}C$ chronologies was removed following a standard correction method (McCarroll and Loader, 2004). The corrected chronologies ($\Delta^{13}C$) particularly showed an enhanced effect of $CO_2$ increase particularly on the deciduous trees (Fig. S6). The current regional annual radiative forcing due to $CO_2$ is about 1.7 Wm$^{-2}$, which is almost fivefold to that of early 20$^{th}$ century. A peak in the transport of dust and greenhouse gases from the Indo-Gangetic plain occurs during the pre-monsoon season, and we anticipate enhanced seasonal $CO_2$ radiative forcing during this critical transitional phase.

Climate sensitivity studies suggest that surface temperature change is higher for aerosols than for greenhouse gases due to its strong longwave effect (Chakraborty and Lee, 2019). The severe lack of aerosol data precludes drawing of any conclusions on black carbon induced ice/snow melting in the Himalaya. Some space-borne studies utilizing ground radiometric measurements indicate that absorbing aerosols have favoured a localized warming over the western-central Himalaya (0.26 ±



0.09 °C/decade) that significantly exceeds the entire Himalaya-Hindu Kush region (Gautam et al., 2010). Relevant studies from Tibetan glaciers suggest that black carbon aerosol deposition is a significant factor contributing to the observed rapid glacier retreat (Xu et al., 2009; Zhao et al., 2017).

To get an estimate of radiative forcing due to ambient concentrations of black carbon aerosols over glacier environment, we utilized data of two existing aethalometer stations from our study region (Fig. 1; Gangotri glacier: 30° 58' 48" N, 79° 4' 48"

E, 3600m asl, Negi et al., 2019; and Dokriani (DOK) glacier: 30°51'6.86" N, 78°45'8.55" E, 3900 m asl). Annual mean radiative forcing (at atmosphere) due to black carbon aerosols (computed as per Dumka et al., 2016) was $+10.1 \pm 3.0$ Wm$^{-2}$ for the Dokriani glacier, $+7.4 \pm 2.2$ Wm$^{-2}$ for the Gangotri glacier, respectively. The lower value for the Gangotri glacier could be due to its location on the leeward side. In support of these results, we found similar mean annual values and diurnal to seasonal dynamics in aerosol radiative forcing with that of sites on the southeastern TP (Zhao et al., 2017; Li et al., 2018;

Zhang et al., 2017). Although we were not able to compute seasonal radiative forcing arising out of black carbon aerosols and/or dust, we anticipate many-fold enhanced pre-monsoonal radiative forcing. These observations indicate the significant contribution of the local forcing factors associated with anthropogenic climate change.

## 4   Conclusions

We present a 273 year-long ice mass loss record from the Ganga basin in the central Himalaya, which is the only record of

annual mass balance variability from the Himalaya so far. Utilizing coherent analyses of tree-ring stable isotope chronologies of regionally dominant tree species and syntheses of regional isotope chronologies, we show that ice mass loss in the basin has accelerated since the mid-20th century to its highest levels of the past 273 years. Stable isotope chronologies of three species of diverse plant functional types and well calibrated $\delta^{13}$C-derived mass balance reconstruction indicates a mean rate of mass loss of $-0.437 \pm 0.025$ m w.e. year$^{-1}$ since 1960. The reconstructed mass balance reveals three major

phases. Mass balance remained positive prior to 1860-70s. Slightly negative but stable mass balance up to 1960s, and a highly negative mass balance since then. Further, our results suggest a doubling of average ice mass loss rate ($-0.577 \pm 0.021$ m w.e. year$^{-1}$) in the last thirty years as compared to 1960-1985 ($-0.275 \pm 0.022$ m w.e. year$^{-1}$). Isotopic and climate coherency analyses indicate that reconstruction is consistent with regional climatic variability and indicate a significant influence of the westerlies in this transitional climate zone of the central Himalaya. The current trend in ice mass loss goes

along with increased anthropogenic aerosol and, $CO_2$ loading and with concurrent change in climatological factors, including a decline of the westerlies and the strength of the Arabian Sea Branch of the Indian summer monsoon. These results present an observational support to calibrate and validate coupled regional climate-glacier models. Improvements in glacier mass-change assessments are still possible and necessary. We suggest that incorporating growth increments of shrubs growing in the alpine zone could be complemented to provide robust estimates of glacier response to future climate scenarios and

modelling studies of glacier contributions to regional runoff.



**Data and materials availability**: All data needed to evaluate the conclusions are present in the paper and/or the Supplementary Material. Any additional data/code related to this paper may be requested from the authors.

**Author contributions**: N.S. & J.S. conceived and designed this study. N.S., J.S., A.B. & C.M. generated tree-ring isotope data. N.S. analyzed final dataset and wrote first draft of the manuscript. J.S. and M.S. performed reconstruction, provided climate datasets and respective analyses. All authors contributed equally to interpretation; discussion and editing of the manuscript.

**Competing interests**: All authors declare no competing interests.

**Acknowledgements**: This work was supported by the Department of Science and Technology (DST) through the 'Centre for Glaciology (CFG) at Wadia Institute of Himalayan Geology (WIHG)'. We thank I. Burchardt, A. Beyer, and R. Höfner-Stich for isotope analyses at FAU Erlangen-Nuremberg. Staffs of the WIHG and CFG (D.P. Dobhal, Aparna Shukla, Indira Karakoti, Pankaj Chauhan and P.K. Garg) are duly acknowledged for discussion and their respective contributions. Vikram Sharma is acknowledged for providing location figure. N.S. acknowledges DST for support under Fast-track young scientist fellowship (File No. SR/FTP/ES-166/2014). M.S. expresses gratitude to Dr. Vandana Prasad (Director, BSIP) and acknowledges Birbal Sahni Research Associate (BSRA) fellowship.

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
