# Peer review of "Central Himalayan tree-ring isotopes reveal increasing regional heterogeneity and enhancement in ice-mass loss since the 1960s"

_The Cryosphere, 2020_

## Referee Comment (RC1) · Anonymous Referee #1 · 5 Aug 2020

General comments: This paper of Singh et al. reconstructed and analyzed the glacier mass balance since 1743 in central Himalaya, using tree ring carbon isotope, it is meaningful for the understanding the glacier variation in Himalaya area. However, the present version has many problems in the results and structure. One of the most important is the glacier mass balance reconstruction. Firstly, is it available to use tree ring carbon to reconstruct glacier mass? The tree ring carbon is influenced by climate, environment and plant physiological factors, the authors should give reasonable explanation about the mechanisms. Secondly, the record of glacier mass balance in only 23 years, also with a few years gap, it is too short and lead to a very low degree of freedom, and then the correlation coefficients between tree ring carbon and glacier mass

balance are not so high. Therefore, I deeply doubt the statistics of the reconstruction. I think the authors should firstly determine the dominant factor of glacier mass, temperature or precipitation in which season? After that, clarify the influence factor of tree ring carbon. Only make these explicit can discuss if tree ring carbon is available to reconstruct glacier mass. Besides, the studied glacier type is not obvious, the authors should give the monthly ice mass, as well as the percentage of summer rainfall and winter snowfall.

Special comments:

Introduction: in the fifth paragraph of this section, you present that tree ring width , you should give some investigation that why tree ring oxygen and carbon could record the information of glacier mass balance. To my knowledge, the climate signal recorded in tree ring carbon isotope is more complicated. Line 174: Do you mean the two conifer trees contain winter-time climate record? what is the growth period of the two conifer trees and the deciduous trees? Line 181: the amount of cellulose is too much, why? Line 165-173: this paragraph should be removed, it should be given in Results section. Section 3: this section should present the carbon series firstly, and then the correlation analysis, finally the reconstruction. The authors should give the raw and corrected 13C values of the studied five trees, as well as the EPS, Rbar and the number of trees at different period.

Line 236-239: the correlation coefficients between 13Cof two PFTs and annual mass balance are not very high, especially in consideration of the low value of degree of freedom (n = 23), so I deeply doubt the reliability of the reconstruction. Line 250-253: this sentence is obscure, you mentioned that the uncertainty still exists in the reconstruction, how does the 13C and the combination of two conifer species can help to minimize several factors. . . . . . . . . . . . . .? Why? Line 262: please give the period of the last glacial advance. Tree ring 13C is affected not only by climate variation, but also the plant physiology and ecological factors, such as the photosynthesis, respiration, soil etc. in term of this, may be tree ring oxygen isotope is better for the reconstruction.

[Figure]

You should give sufficient discussion on the impact factors of tree ring carbon isotope, the physical mechanisms between tree ring 13C and glacier mass balance must be discussed, the basis of the statistical correlation between them is not persuasive, Line 370: please give the reference. Line 376-377: please give the reference. Table S1: The first order autocorrelation is very high, why? Section 3.3: this section is too complicated and have no focus. Line 236-240: what is the relationship between 13C of both evergreen conifer species, from Figure S6, the difference between two series is obvious, I don't think it is available to combine the two series.

---

## Referee Comment (RC2) · Anonymous Referee #2 · 23 Sep 2020

Dear Editor,

Thank you for the opportunity to review the manuscript, titled "Central Himalayan tree-ring isotopes reveal increasing regional heterogeneity and enhancement in ice-mass loss since the 1960s." I enjoyed reading the manuscript. The manuscript presents some exciting results. There are several statistical results presented that show the depth of the analysis carried out by the authors. However, I have several minor comments which the authors must address. Specifically, the figures in the main text need substantial modifications. Please kindly see my comments that are provided below and also throughout the main text pdf. Once these comments are addressed, the

manuscript should be good for publications. Please kindly let me know if you any further questions.

To the authors: The manuscript, titled "Central Himalayan tree-ring isotopes reveal increasing regional heterogeneity and enhancement in ice-mass loss since the 1960s", tries to address the correlation between $\delta$13C and glacier mass balance and their temporal evolution in the past. The authors provided a detailed description of their tree ring isotope measurements and showed a decent correlation with the reconstructed glacial mass balance for the past 273 years in the central Himalaya. The authors attempted several different statistical tests and presented their results. The results clearly show a shift in climate proxies since 1960's. The supplementary figures were very helpful to assess the results and to understand the in-depth discussion presented in the paper. However, several areas could be improved. Please kindly see my minor comments throughout the main text pdf and the comments attached here as follows.

Other comments: In the introduction, it is not apparent immediately what is the exact study area. What I understood after much reading is that the four major glaciated valleys that are located in the central Himalaya are chosen because of the availability of the mass balance data for the past few decades. The central Himalaya was arbitrarily divided into the western central Himalaya (WCH) and the eastern Central Himalaya (ECH). No such map is presented to delineate what areas the authors mean by WCH and ECH. The authors started their introduction with the "transitional western Himalaya." They presented an arbitrary map of what is shown here as the ISM dominated area, westerlies dominated area, and in between is the transitional area. These areas are not well defined and likely have tremendous overlapping (see my comments below in figure 1). The authors frequently also invoked the Ganges basin. The Ganges basin is enormous and incorporated areas beyond the central Himalaya. I highly recommend that the authors must clarify, first, what area is the central Himalaya in this study and what glaciated valleys constitute ECH and WCH. It must also be clear if any portion of ECH and WCH is part of westerlies/ISM/transitional. The manuscript then must use consistent regions for interpretation. Avoid using the Ganges basin unless results from the entire basin are presented.

So many acronyms were used to the point that it distracts the reader from the smooth reading of the paper. Sometimes the acronyms were used only a handful of times. I recommend only use acronyms that are repeated on several occasions, are commonly used (e.g., SST), and are mentioned in the figures, tables, or equations. In every other case, spell the full acronyms.

Lines 103: The CRU dataset for precipitation for the high-altitude sites has several limitations. This needs to be explained somewhere in the text or supplementary.

Results and discussions must be presented distinctly. It is hard to follow, which is a result and which is a discussion material. This would also make the discussion flow well.

Figure 1: The symbols are hard to read. I would suggest making them slightly bigger and use distinct color-coding to make them visible. Similarly, the numbers are hard to read. Consider making them bold black, or red. I could not see the stars (meteorological stations). The four major glaciers must also need to annotate in the main figure (top panel). The north arrow is missing in Figures B, C, D. The scale is also very small in figure A. All other panels need scale. Why are latitudes absent? Similarly, longitudes are required for A and B. Ideally, the top panel figure must be A, then from left to right (bottom panel), it should B, C, D, E. The figure scale at the top panel is 265 and 530 km. That's very odd. Consider making them round by zoom in or out (e.g., 250, 500 km). Please annotate the name of each glacier in each bottom panel. What are the main tree ring sites studied in this paper (i.e., the new sites)? This needs to be appropriately highlighted, and their symbols must be distinct from the published sites. Finally, what is the source/basis of the dashed regions? It is not clear to me how you defined those regions. Presently there is a strong latitudinal, altitudinal, and longitudinal climate gradient in the orogen. If the paper is based on comparing the tree ring/glacier signals across distinct climatic regions, they must be well defined. I'm afraid I have to disagree that ISM does extend in the NW Himalaya and even in parts of the NW and interior of Tibet. ISM extended further north in the past when it was stronger than the present. Similarly, westerlies also largely influence parts of the central Himalaya in the winter currently shown under ISM dominated. Therefore, the current zonation is vague and needs proper justification.

Figure 2: The x-axis of all the time-series data must be the same for qualitative comparison. It is hard to find the usefulness of the ice core proxies and glacier length change data at its current configuration. Also, note that length response may be affected by glacier size, slope, and hypsometry. A better explanation is required as to why they should be used as a proxy for glacier health. How about changes in the ELAs?

Figure 3: This figure is not well organized at the moment. I recommend organizing them into a single page robust figure or separate figures. Each panel must be designated as A), B) . . . for easier understanding and reading the figure caption (this applies to other figures as well). There is no need to show MB twice in the same plot. Keep the x-axis range and length for all the graphs the same.

Figure S4: I found this figure very important, and if possible, it should be part of the main text.

Please also note the supplement to this comment:
https://tc.copernicus.org/preprints/tc-2020-128/tc-2020-128-RC2-supplement.pdf

———————————————————

[revised manuscript text omitted]

---

## Editor Comment (EC1) · Chris R. Stokes (Editor) · 29 Sep 2020

I would like to thank both reviewers for their constructive comments on this manuscript. Whilst both identify a number of issues (in terms of the content and presentation), they also recognise the importance of the work and I would encourage the authors to consider them carefully and submit a revised manuscript.

---

## Author Comment (AC1) · 27 Oct 2020

General comments: This paper of Singh et al. reconstructed and analyzed the glacier mass balance since 1743 in central Himalaya, using tree ring carbon isotope, it is meaningful for the understanding the glacier variation in Himalaya area. Response: We are thankful to the reviewer for valuable comments and suggestions that improved this manuscript. In this paper, we reconstructed annual variability of four 'benchmark glaciers' of the Uttarakhand Himalaya utilizing tree-ring carbon isotopes of two dominant conifer species growing in the valleys. We also analyzed the variability of tree-ring and ice-core oxygen isotopes on a central Himalayan-scale. Comment: However, the

present version has many problems in the results and structure. One of the most important is the glacier mass balance reconstruction. Firstly, is it available to use tree ring carbon to reconstruct glacier mass? The tree ring carbon is influenced by climate, environment and plant physiological factors, the authors should give reasonable explanation about the mechanisms. Response: In this revised version, we improved this manuscript in light of the reviewer's comments. Here we would like to mention that tree-ring carbon isotopes are strong predictors of local ecohydrology, while $\delta$18O retain physical climatic signals that are essentially coherent over a large region. Studies are available that reconstructed glacier mass balance based on tree-ring carbon isotopes of zonal dominant conifers growing in the glacier valleys. Even seasonal mass balance reconstructions have been attempted utilizing tree-ring parameters and period of growth. Recently, Zhang et al. (2019) reconstructed mass balance of TS. Tuyuksuyskiy glacier in the Tianshan Mountains utilizing the strong link between $\delta$13C variability and growing season temperatures, and the premise that climate is a bridge that indirectly connects mass balance of a glacier with tree growth.

Prevailing climate controls the amount of snow accumulation and ablation, which determine the retreat or advancement of a glacier. Similarly, tree growth (and $\delta$13C composition) is a result of an integrated response to climate. Thus, trees growing in the valleys and glacier mass balance respond to the same climatic forcing. The variability of the 13C/12C ratio ($\delta$13C composition) in tree-ring cellulose is mainly controlled by photosynthesis and stomatal conductance. At dry sites, these processes depend mostly on relative humidity and soil moisture availability, whereas at moist sites the C-isotope ratio reflects the variability of radiation and temperature. Dendroglaciological and paleoclimate studies from the warm-moist, monsoon-dominated Himalaya suggest that temperature changes are the prime factor for glacier fluctuations rather than changes in precipitation. C-isotope ratios in tree-rings of evergreen conifers growing in energy-limited valley environment strongly reflect the temperature as well as some measure of water availability (Figure S3). Therefore, in these environments the factors that influence photosynthesis are strongly correlated with $\delta$13C of tree-rings.

[Figure]

Vegetation growth in the Himalaya is known to be primarily regulated by temperature. Therefore, mass balance reconstruction of valley glaciers based on tree-ring $\delta13C$ assumes significance, particularly in the warm-moist valley environment where coupling between carbon and water cycle is strong (Singh et al., 2014). Moreover, given strong land-atmosphere interaction (Singh et al., 2019; Tuinenburg et. al., 2012), ecohydrological memory (Chauhan and Ghosh, 2020), and vegetation-regulated moisture recycling (Keys et al., 2016), tree-ring $\delta13C$ is a strong proxy to study local valley-scale climate. In addition, remarkably high interspecies and spatiotemporal coherence observed between tree-ring $\delta13C$ and $\delta18O$ probably indicate a high ecophysiological-ecohydological coupling, particularly for the studied region (Figure 3b), increasing the reliability of the reconstruction. To conclude, in humid environments, the $\delta13C$ composition of tree-rings is determined by the photosynthesis rate, which in turn depends upon temperature. Higher growing season temperatures stimulate photosynthesis rates. This facilitates stable isotope fractionation process, resulting in decreased intercellular $CO2$ concentration and fractionation against the heavier 13C isotope, which leads to increased higher $\delta13C$. Thus, climate (temperature) is a bridge that indirectly connects the mass balance of a glacier with tree-ring C isotope ratios. Hence, the mass balance can be reconstructed using stable carbon isotope chronologies from trees growing in the proximity of the glacier.

Comment: Secondly, the record of glacier mass balance in only 23 years, also with a few years gap, it is too short and lead to a very low degree of freedom, and then the correlation coefficients between tree ring carbon and glacier mass balance are not so high. Therefore, I deeply doubt the statistics of the reconstruction. Response: Thank you for your concern. We admit that due to inclement weather conditions, difficult terrain, and consequent logistic-infrastructural reasons, the longest available record of glacier mass balance is maximum upto 23 years for the Uttarakhand Himalaya. However, to overcome the limitation of short glacier mass balance data, we used the well-established and robust statistical leave-one-out cross-validation method (Michaelsen, 1987). This statistical approach has often been applied under similar

limitations of short calibration data length in dendroclimatic studies, including the Himalayan region (Shah et al., 2013; Shekhar et al., 2017; Yadav and Bhutiyani, 2013, Duan et al., 2013; Zhang et al., 2019). The leave-one-out method (Michaelsen, 1987) fits a regression model to all the data except for a single point, and then makes a prediction at that point. The estimates for every data point form a modelled time series, which is compared with the actual values. Our mass balance reconstruction captured robust significant verification statistics (Table S3, Figure S4), thereby validating the regression model. In addition, our mass balance reconstruction passed other verification statistics. Particularly, the sign-test and the root mean square error (RMSE) have been widely used as a validation parameter for the reliability of climate reconstructions (Cook and Kairiukstis 1990). We recorded a small RMSE value (0.166), indicating a small error in the reconstructed values. The low Durbin-Watson coefficient (1.21), and the F-test value (12.37) in the verification also indicate the significance of the regression model. The positive values of reduction of error (RE) and coefficient of efficiency (CE) underpins significant skills in the reconstruction and acceptable model performance (Cook et al. 1999) (Table S3). Comment: I think the authors should firstly determine the dominant factor of glacier mass, temperature or precipitation in which season? After that, clarify the influence factor of tree ring carbon. Only make these explicit can discuss if tree ring carbon is available to reconstruct glacier mass. Response: We appreciate the reviewer's suggestions. Dendroglaciological and paleoclimate studies from the region suggest that temperature change is the prime factor for glacier fluctuations on a decadal timescale rather than changes in precipitation. In the present study, correlation analysis with CRU temperature data indicate a high coherence with mass balance dynamics (r = -0.78, P < 0.001). In contrast, correlation with gridded precipitation including that of northern India and ISM rainfall indicate a low association (Figure 3a). While the response function analyses (Figure S3) indicate that tree-ring $\delta$13C composition is primarily influenced by temperature. Thus, our analyses indicate that temperature is a main governing parameter of glacier mass balance and tree-ring $\delta$13C composition in the region. Comment: Besides, the studied glacier type is not

obvious, the authors should give the monthly ice mass, as well as the percentage of summer rainfall and winter snowfall. Response: Thank you for your comment. The glacier type and its geomorphological characteristics including hypsometric curves have been detailed in section 2.2. The studied glacier belongs to land-terminated valley-type glaciers of simple configuration and hypsometry, which are most suitable to infer paleoclimatic information and are therefore designated by workers as 'benchmark glaciers' of the humid central Himalaya. As mentioned previously, due to obvious logistical reasons, no data on monthly ice mass values for the Himalayan glaciers are available. Based on the analyses of available meteorological records, we indicated that warm-wet summer months receive about 80% of the mean annual precipitation, while winter snowfall contributes the rest. This observation is supported by many previous studies. Special comments: Comment: Introduction: in the fifth paragraph of this section, you present that tree ring width, you should give some investigation that why tree ring oxygen and carbon could record the information of glacier mass balance. To my knowledge, the climate signal recorded in tree ring carbon isotope is more complicated. Response: We appreciate the reviewer's comment. As suggested, we have now incorporated (Lines: 82-86) the linking mechanism and the basic premise explaining why tree-ring carbon isotope ratios could record the information of glacier mass balance. Comment: Line 174: Do you mean the two conifer trees contain winter-time climate record? what is the growth period of the two conifer trees and the deciduous trees? Response: Early wood accounts for most of the whole ring in coniferous species and mainly carries a climate signal of the previous autumn or winter and early-spring seasons (Zeng et al., 2017). Conifer trees mostly use the snowmelt water of the previous winter for wood formation in the early growing season. The two conifer species are evergreen, while growth of the deciduous trees occurs between April and September in the Himalaya. Comment: Line 181: the amount of cellulose is too much, why? Response: The amount of cellulose utilized for IRMS analyses is per common isotope-dendrochronological procedure. Comment: Line 165-173: this paragraph should be removed, it should be given in Results section. Response: Thank you for

the suggestion. We modified as suggested. Now lines: 221-263. Comment: Section 3: this section should present the carbon series firstly, and then the correlation analysis, finally the reconstruction. The authors should give the raw and corrected 13C values of the studied five trees, as well as the EPS, Rbar and the number of trees at different period. Response: We modified as suggested (Lines: 221-263). Raw and corrected 13C values of the studied species are presented in Figure S6. Generally, EPS > 0.85 and Rbar statistics are used in tree-ring studies for determining the reliable length of chronologies in dendroclimatology, which commonly lose replication when moving further into the past (Yadav and Bhutiyani, 2013). However, in case of dendro-isotope studies (because of the high coherency of isotope series from individual trees) the common approach is to combine more than 3-4 tree cores over the time, which are sufficient for establishing a reliable tree-ring isotope chronology (Zeng et al., 2017; Singh et al., 2019; Sano et al., 2012, 2013, 2017; Xu et al., 2018). In the present study, we have used a chronology with 5 tree cores over the entire series of respective species (i.e., 1743-2015 in case of Abies and 1920-2015 in Picea). Comment: Line 236-239: the correlation coefficients between 13Cof two PFTs and annual mass balance are not very high, especially in consideration of the low value of degree of freedom (n = 23), so I deeply doubt the reliability of the reconstruction. Response: In paleoclimate reconstructions, more than 35% variance explained in the calibration period is acceptable to tell the past history, although a higher variance explained is preferred (Fritts 1976). In our reconstruction, we achieved 46.5% variance explained during the calibration period, which is quite high and sufficient for a reconstruction. Further, our regression model was rigorously verified with multiple verification statistics (please see Table S3). Detailed statistics, previous studies, and the reliability of recon- struction have been discussed in our response to earlier comments (see Table S3). Comment: Line 250-253: this sentence is obscure, you mentioned that the uncertainty still exists in the reconstruction, how does the 13C and the combination of two conifer species can help to minimize several factors. . . . . . . . . . . . .? Why? Response: Thank you for pointing to this unclear expression. From these lines (now lines: 257-259), we

Interactive
comment

contend that the short observation period might create a certain level of uncertainty during pre-observation period. The use of highly sensitive isotope chronologies (with respect to tree-ring width) could avoid or minimize uncertainty that arises due to sensitivity issues associated with ring width. Thus, the observed high coherence in the isotope chronologies of two conifer species allowing their combination, could help to minimize the both low and high frequency noise and enhance the climate signal. Comment: Line 262: please give the period of the last glacial advance. Response: Incorporated in the text as suggested (Line: 272). Comment: Tree ring 13C is affected not only by climate variation, but also the plant physiology and ecological factors, such as the photosynthesis, respiration, soil etc. in term of this, may be tree ring oxygen isotope is better for the reconstruction. Response: The sensitivity of tree-ring $\delta$13C to climate is strictly controlled by the environmental conditions impacting tree physiology. Tree-ring $\delta$13C composition is mainly controlled by photosynthesis and stomatal conductance. At warm-moist sites such as in the Himalaya, the C-isotope ratio should reflect the variability of irradiance and temperature. In contrast, tree-ring $\delta$18O consistently reflects variations of the source water $\delta$18O under humid conditions. Further, leaf-water enrichment occurring during transpiration usually modifies the oxygen signal contained in the source water. Therefore, we found a strong correlation with atmospheric moisture content, while no relation was observed with the glacier mass balance. Comment: You should give sufficient discussion on the impact factors of tree ring carbon isotope, the physical mechanisms between tree ring 13C and glacier mass balance must be discussed, the basis of the statistical correlation between them is not persuasive. Response: Thank you for your suggestions. We discussed this point as suggested in response to an earlier comment, while we also have introduced (Lines: 82-86) the linking mechanism and the basic premise why tree-ring carbon isotopes can record information on glacier mass balance. Comment: Line 370: please give the reference. Response: Reference added as suggested (Line: 380). Comment: Line 376-377: please give the reference. Response: Incorporated as suggested (Line: 387). Comment: Table S1: The first order autocorrelation is very high, why? Response: The high autocorrelation suggests the presence of carbon carry-over effects and a memory effect. Comment: Section 3.3: this section is too complicated and have no focus. Response: In this section utilizing six tree-ring $\delta$18O chronologies from the central Himalaya, we discussed the mechanisms of increasing climate heterogeneity since the mid-twentieth century between the western (WCH) and eastern (ECH) parts of the central Himalaya. This section discussed that temperature is the prime climatic factor that influences the behavior of mass balance at interannual to multi-decadal time scales. We further discussed the implications of large-scale changes in atmospheric circulation on cryospheric mass balance in the central Himalaya. We showed that over the WCH (compared to ECH), the westerlies and the Arabian Sea branch of ISM has a significant impact on annual mass balance behavior. Using spectral wavelet coherence analyses, we show a lack of strong coherence in the high frequency bands between mass balance dynamics and SST indices (ENSO Modoki, IOD and PDO). We feel that this information is relevant in the context of the major aims of the paper. Comment: Line 236-240: what is the relationship between 13C of both evergreen conifer species, from Figure S6, the difference between two series is obvious, I don't think it is available to combine the two series. Response: Both conifer species (Abies pindrow and Picea smithiana) revealed a high inter annual correlation (r = 0.84, P < 0.001; please see Table S1 and S4), a basis to merge both chronologies together. In doing so, we strengthen the common (climatic) signal in the final isotope chronology and dampen possible species-specific individual physiological reactions to local factors. Additionally, the mean difference (‰ between the two conifer species is only 0.53 ‰ $whereas the mean difference between conifer and deciduous species is 1.6 - -2.2 (Table S1 and S4). Reference : Chauhan, T., and Ghosh, S., 2020. Partitioning of memory and real- time connections between variables in Himalayan ecohydrological processes networks. Journal of Hydrology, doi : https : //doi.org/10.1016/j.hydrol.2020.125434 Keys, P.W., Wang - Erlandsson, L. and Gordon, L.J., 2016. Revealing invisible water : moisture recycling as an ecosystem service. PloS one, 11(3), p.e0151993. Singh, N., Patel, N.R., Bhattacharya, B.K., Soni, P., linkages of carbon and water fluxes in subtropical pine (Pinus roxburghii) ecosystem. Agricultural and forest meteorology, 197,$

218. *Singh, N., Singhal M., Chhikara, S., Karakoti, I., Chauhan, P., Dobhal, D.P., 2019. Radiation and energy balance dynam*

420. *Tuinenburg, O.A., Hutjes, R.W.A. and Kabat, P., 2012. The fate of evaporated water from the Ganges basin. Journal of Ge*
*Atmospheres, 117(D1).*

*Please also note the supplement to this comment* :
*https* : *//tc.copernicus.org/preprints/tc − 2020 − 128/tc − 2020 − 128 − AC1 −*
*supplement.pdf*

---

## Author Comment (AC2) · 27 Oct 2020

Comments to the authors: The manuscript, titled "Central Himalayan tree-ring isotopes reveal increasing regional heterogeneity and enhancement in ice-mass loss since the 1960s", tries to address the correlation between $\delta$13C and glacier mass balance and their temporal evolution in the past. The authors provided a detailed description of their tree ring isotope measurements and showed a decent correlation with the reconstructed glacial mass balance for the past 273 years in the central Himalaya. The authors attempted several different statistical tests and presented their results. The results clearly show a shift in climate proxies since 1960's. The supplementary figures

were very helpful to assess the results and to understand the in-depth discussion presented in the paper. However, several areas could be improved. Please kindly see my minor comments throughout the main text pdf and the comments attached here as follows. Response: We are grateful to the reviewer for the appreciation, valuable comments and suggestions that improved the quality of this manuscript. We have now revised this manuscript in light of the comments and suggestions. Other comments: Comment: In the introduction, it is not apparent immediately what is the exact study area. What I understood after much reading is that the four major glaciated valleys that are located in the central Himalaya are chosen because of the availability of the mass balance data for the past few decades. Response: Thank you for pointing out that the study area was not clearly described. As suggested, we now introduced the study area in this section (Lines: 51-61). Comment: The central Himalaya was arbitrarily divided into the western central Himalaya (WCH) and the eastern Central Himalaya (ECH). No such map is presented to delineate what areas the authors mean by WCH and ECH. The authors started their introduction with the "transitional western Himalaya." They presented an arbitrary map of what is shown here as the ISM dominated area, westerlies dominated area, and in between is the transitional area. These areas are not well defined and likely have tremendous overlapping (see my comments below in figure 1). Response: The central Himalaya was divided into the western central Himalaya (WCH) and the eastern Central Himalaya (ECH), based on the existing records of tree-ring $\delta18O$ from the region (from the Manali to Bhutan: Figure 1). Previous tree-ring isotope studies (Sano et al., 2012, 2013, 2017; Singh et al., 2019; Xu et al., 2018) as well as our analyses on the response of tree-ring $\delta18O$ records to the physical climate of the region formed the basis for the division into WCH and ECH (Please see Figure 3b and Table S6). We now presented an entirely re-designed map (Figure 1) to delineate WCH and ECH and to respond to all comments related to figure 1. A high correlation between tree-ring sites from Manali to Jageswar (WCH) may be noted, whereas correlation decline sharply towards Bhutan (Table S6). Our presentation of the map (Figure 1) showing ISM-, Westerlies-dominated area and

transitional area is based on Huang et al. (2019). However, the reviewer has a point that these areas are likely to have tremendous overlapping. Therefore, in our modified map (Figure 1), we have removed these distinctions. Comment: The authors frequently also invoked the Ganges basin. The Ganges basin is enormous and incorporated areas beyond the central Himalaya. I highly recommend that the authors must clarify, first, what area is the central Himalaya in this study and what glaciated valleys constitute ECH and WCH. It must also be clear if any portion of ECH and WCH is part of westerlies/ISM/transitional. The manuscript then must use consistent regions for interpretation. Avoid using the Ganges basin unless results from the entire basin are presented. Response: Thank you for pointing this out. We have now avoided using the term 'Ganges basin'; instead we now used 'Uttarakhand Himalaya'. As suggested, we indicated that (Lines: 93-95, Figure 1) the six tree-ring sites distributed across the region from Manali (at the northwestern periphery of ISM incursions; Sano et al., 2017) upto Bhutan constitute the central Himalaya. The four studied glaciated valleys of the Uttarakhand Himalaya (Dokriani, Chorabari, Tipra Bamak and Dunagiri) constituted WCH, while the region around the Dasuopu ice-core site to Bhutan is ECH. The manuscript used these regions for interpretation. Comment: So many acronyms were used to the point that it distracts the reader from the smooth reading of the paper. Sometimes the acronyms were used only a handful of times. I recommend only use acronyms that are repeated on several occasions, are commonly used (e.g., SST), and are mentioned in the figures, tables, or equations. In every other case, spell the full acronyms. Response: As suggested, we have now taken care of the acronyms and their use. Comment: Lines 103: The CRU dataset for precipitation for the high-altitude sites has several limitations. This needs to be explained somewhere in the text or supplementary. Response: We agree with the reviewer's point of view that the CRU dataset for precipitation, particularly for high-altitude regions has limitations. However, previous studies from the region comparing in-situ precipitation datasets and CRU precipitation obtained significant relations (Shekhar et al., 2018; Yadav et al., 2014, 2015). Therefore, we have complemented CRU precipitation with datasets from

meteorological stations (utilized by Singh et al., 2019). This has now been indicated in the text (Lines: 110-112). Comment: Results and discussions must be presented distinctly. It is hard to follow, which is a result and which is a discussion material. This would also make the discussion flow well. Response: Since our results are presented in four distinct sections (mass balance reconstruction, phases in the mass balance dynamics, regional climate heterogeneity, and local forcing factors), it was easier and more appropriate to discuss the results section-wise and separately. However, in response to the reviewers' suggestion (#1 and #2), we tried to write a respective discussion in flow and presented the results as succinctly as possible. Comment: Figure 1: The symbols are hard to read. I would suggest making them slightly bigger and use distinct color-coding to make them visible. Similarly, the numbers are hard to read. Consider making them bold black, or red. I could not see the stars (meteorological stations). The four major glaciers must also need to annotate in the main figure (top panel). The north arrow is missing in Figures B, C, D. The scale is also very small in figure A. All other panels need scale. Why are latitudes absent? Similarly, longitudes are required for A and B. Ideally, the top panel figure must be A, then from left to right (bottom panel), it should B, C, D, E. The figure scale at the top panel is 265 and 530 km. That's very odd. Consider making them round by zoom in or out (e.g., 250, 500 km). Please annotate the name of each glacier in each bottom panel. What are the main tree ring sites studied in this paper (i.e., the new sites)? This needs to be appropriately highlighted, and their symbols must be distinct from the published sites. Finally, what is the source/basis of the dashed regions? It is not clear to me how you defined those regions. Presently there is a strong latitudinal, altitudinal, and longitudinal climate gradient in the orogen. If the paper is based on comparing the tree ring/glacier signals across distinct climatic regions, they must be well defined. I'm afraid I have to disagree that ISM does extend in the NW Himalaya and even in parts of the NW and interior of Tibet. ISM extended further north in the past when it was stronger than the present. Similarly, westerlies also largely influence parts of the central Himalaya in the winter currently shown under ISM dominated. Therefore,

the current zonation is vague and needs proper justification. Response: In response to critical comments related to Figure 1, we have now entirely revised this figure in addition to the lower panels (now B, C, D and E). We enlarged the symbols and used a distinct color-coding. The numbers have been made clearer. The meteorological stations (stars) have been made distinct. The four glaciers have been annotated distinctly (top panel). In lower panel figures (now B, C, D and E), the name of glaciers, north arrow, scale, latitudes and longitudes have been indicated. The figure scale in top panel (A) has been modified to 250 and 500 km. Now, the new tree-ring site and the published sites have been highlighted appropriately. The source of the dashed regions (ISM, westerlies dominated area and transitional area) is Huang et al. (2019). However, the reviewer has a good point that these areas are likely to have tremendous overlapping. Therefore, in our modified map (Figure 1), we have removed these zonations. Comment: Figure 2: The x-axis of all the time-series data must be the same for qualitative comparison. It is hard to find the usefulness of the ice core proxies and glacier length change data at its current configuration. Also, note that length response may be affected by glacier size, slope, and hypsometry. A better explanation is required as to why they should be used as a proxy for glacier health. How about changes in the ELAs? Response: The reviewer appropriately noted that the length response may be affected by various geomorphological factors and the time-series data must be the same for a comparison. Therefore, we have now deleted this glacier length change data (which has been adopted after Bolch et al., 2012). Comment: Figure 3: This figure is not well organized at the moment. I recommend organizing them into a single page robust figure or separate figures. Each panel must be designated as A), B).... for easier understanding and reading the figure caption (this applies to other figures as well). There is no need to show MB twice in the same plot. Keep the x-axis range and length for all the graphs the same. Response: Figure 3 as well as the rest of the figures have been modified as suggested. Comment: Figure S4: I found this figure very important, and if possible, it should be part of the main text. Response: Figure S4 describes the three distinct results, viz., (1) correlation matrix and pair plots

for $\delta$13C chronologies of different plant functional types, (2) relationship of mean $\delta$13C conifer chronologies and observed mass balance, and (3) comparison of observed and reconstructed mass balance. Therefore, to club these distinct results, we suggest that supplementary material would be more appropriate. Comment: Please also note the supplement to this comment: https://tc.copernicus.org/preprints/tc-2020-128/tc-2020-128-RC2-supplement.pdf Response: We thank the reviewer for a patient reading and in-depth comments. We have now addressed each and every supplementary comment in the main text. Reference: Huang, et al. (2019). Temperature signals in tree-ring oxygen isotope series from the northern slope of the Himalaya. Earth and Planetary Science Letters, 506, 455–465. Shekhar, et al. (2018). Tree-ring based reconstruction of winter drought since 1767 CE from Uttarakashi, Western Himalaya. Quaternary International, 479, 58-69. Yadav, et al. (2014). Premonsoon precipitation variability in Kumaun Himalaya, India over a perspective of ∼300 years. Quaternary International, 325, 213-219. Yadav, et al. (2015). Tree-ring footprints of drought variability in last ∼300 years over Kumaun Himalaya, India and its relationship with crop productivity. Quaternary Science Reviews, 117, 113-123.

Please also note the supplement to this comment:
https://tc.copernicus.org/preprints/tc-2020-128/tc-2020-128-AC2-supplement.pdf

**Supplement:**

[revised manuscript text omitted]